# Yeasts Have Evolved Divergent Enzyme Strategies To Deconstruct and Metabolize Xylan

Jonas L. Ravn,[a] Amanda Sörensen Ristinmaa,[a] Tom Coleman,[a] Johan Larsbrink,[a,b] Cecilia Geijer[a]

aDepartment of Life Sciences, Chalmers University of Technology, Gothenburg, Sweden
bWallenberg Wood Science Center, Chalmers University of Technology, Gothenburg, Sweden

**ABSTRACT** Together with bacteria and filamentous fungi, yeasts actively take part in the global carbon cycle. Over 100 yeast species have been shown to grow on the major plant polysaccharide xylan, which requires an arsenal of carbohydrate active enzymes. However, which enzymatic strategies yeasts use to deconstruct xylan and what specific biological roles they play in its conversion remain unclear. In fact, genome analyses reveal that many xylan-metabolizing yeasts lack expected xylanolytic enzymes. Guided by bioinformatics, we have here selected three xylan-metabolizing ascomycetous yeasts for in-depth characterization of growth behavior and xylanolytic enzymes. The savanna soil yeast *Blastobotrys mokoenaii* displays superior growth on xylan thanks to an efficient secreted glycoside hydrolase family 11 (GH11) xylanase; solving its crystal structure revealed a high similarity to xylanases from filamentous fungi. The termite gut-associated *Scheffersomyces lignosus*, in contrast grows more slowly, and its xylanase activity was found to be mainly cell surface-associated. The wood-isolated *Wickerhamomyces canadensis*, surprisingly, could not utilize xylan as the sole carbon source without the addition of xylooligosaccharides or exogenous xylanases or even co-culturing with *B. mokoenaii*, suggesting that *W. canadensis* relies on initial xylan hydrolysis by neighboring cells. Furthermore, our characterization of a novel *W. canadensis* GH5 subfamily 49 (GH5_49) xylanase represents the first demonstrated activity in this subfamily. Our collective results provide new information on the variable xylanolytic systems evolved by yeasts and their potential roles in natural carbohydrate conversion.

**IMPORTANCE** Microbes that take part in the degradation of the polysaccharide xylan, the major hemicellulose component in plant biomass, are equipped with specialized enzyme machineries to hydrolyze the polymer into monosaccharides for further metabolism. However, despite being found in virtually every habitat, little is known of how yeasts break down and metabolize xylan and what biological role they may play in its turnover in nature. Here, we have explored the enzymatic xylan deconstruction strategies of three underexplored yeasts from diverse environments, *Blastobotrys mokoenaii* from soil, *Scheffersomyces lignosus* from insect guts, and *Wickerhamomyces canadensis* from trees, and we show that each species has a distinct behavior regarding xylan conversion. These findings may be of high relevance for future design and development of microbial cell factories and biorefineries utilizing renewable plant biomass.

**KEYWORDS** yeast, xylan, xylanase, CAZymes, microbial co-cultures

Yeasts inhabit every biome in nature and live in a variety of ecological niches, including soil, water, air, and plant and fruit surfaces (1). Together with filamentous fungi and bacteria, yeasts actively partake in polysaccharide decomposition in decaying plant biomass but have been much less studied. In a recent review of cellulose and xylan-degrading yeasts, we concluded that cellulose-degrading yeasts are rare, whereas xylan-degrading yeasts are rather widespread in nature. In fact, more than 100 yeast

Address correspondence to Cecilia Geijer, cecilia.geijer@chalmers.se, or Johan Larsbrink, johan.larsbrink@chalmers.se.

The authors declare no conflict of interest.

species have been identified to date that display xylanolytic capacities, but their precise xylan degradation strategies and biological role(s) in the ecosystem remain largely unexplored (2).

Microorganisms use carbohydrate active enzymes (CAZymes), collected and described within the CAZy database (www.cazy.org; 3), to degrade xylan polymers into monosaccharides that can be further catabolized and used as carbon and energy sources. Xylans comprise a backbone of $\beta$-1,4-linked D-xylose residues that may be highly O-acylated and substituted by $\alpha$-1,2-linked (methyl)-glucuronic acid, $\alpha$-1,2- or $\alpha$-1,3-linked arabinosyl units, and phenolic compounds (4, 5). These decorations give rise to different types of xylans, typically grouped into arabinoxylan (AX), glucuronoxylan (GX), and glucuronoarabinoxylan (GAX) (6). Xylan degradation requires an arsenal of CAZymes such as carbohydrate esterases (CEs) and glycoside hydrolases (GHs), with the most common xylan backbone-cleaving endo-$\beta$-1,4-xylanases being found in glycoside hydrolase family 10 (GH10), GH11, and GH30 (7, 8).

In a previous study, we predicted CAZymes in 332 genome-sequenced ascomycetous yeasts (9) and identified several new xylanolytic species (10). Combined with additional sequenced xylanolytic yeasts in the literature, 24 species and their respective CAZymes were bioinformatically mapped (2). Interestingly, we identified different subgroups of xylanolytic yeasts regarding their putative endo-$\beta$-1,4-xylanases. Only a single species possesses a GH11 xylanase, while GH10 and GH30_7 enzymes were encoded by five and three species, respectively. However, 18 out of the 24 yeasts lack enzymes from those common xylanase families, indicating that they possess novel xylanases or xylanolytic strategies (2, 10). Many possess GH5 enzymes from subfamilies without known specificities, and we hypothesize that these may be missing pieces in the xylan deconstruction puzzle of yeasts (Fig. 1).

In this study, we characterized the xylanolytic strategies of three phylogenetically diverse yeasts with different CAZyme profiles (10): *Blastobotrys mokoenaii* (phylogenetically in the *Trichomonascaceae* clade; isolated from savanna soil), which possesses a GH11 and a GH30_7 xylanase, *Scheffersomyces lignosus* (CUG-Ser1 clade; isolated from the gut of wood-boring insects), which possesses a GH10 xylanase, and *Wickerhamomyces canadensis* (*Phaffomycetaceae* clade; isolated from Canadian red pine), which lacks any obvious endo-$\beta$-1,4-xylanases. We successfully applied a strategy based on a rational bioinformatic selection of enzyme targets, heterologous production, and screening of recombinant enzymes on a wide diversity of carbohydrates, which confirmed xylanase activities of a GH11 enzyme from *B. mokoenaii* and a GH10 enzyme from *S. lignosus*. Moreover, we demonstrated that *W. canadensis* possesses functional GH5 subfamily 49 (GH5_49) and GH5_22 endo-xylanases.

## RESULTS

**Yeast xylan growth and subcellular localization of xylanolytic activities.** To characterize the xylanolytic growth behavior of *B. mokoenaii*, *S. lignosus*, and *W. canadensis*, we monitored their growth in minimal medium with beechwood GX or xylose as sole carbon sources. *B. mokoenaii* reached a relatively high final optical density at 600 nm ($OD_{600}$; ~14) on beechwood GX with similar growth characteristics as on xylose. In contrast to the two other species with typical spherical yeast shapes (Fig. 2E and F), *B. mokoenaii* displayed pseudomycelia (Fig. 2D) and multilateral budding morphology forming hyphae and blastoconidia with elongated setae (11, 12). *S. lignosus* displayed an extended lag phase and slow growth on beechwood GX compared to xylose, while no growth of *W. canadensis* on xylan could be detected within the time frame of the experiment (96 h). We hypothesized that the lack of growth may be attributed to a lack of sensing the carbohydrate polymer and decided to supplement the medium with 0.2% xylooligosaccharides (XOs), which enabled *W. canadensis* to grow to an $OD_{600}$ of ~5. As 0.2% XOs alone did not support significant growth (OD, ~1), this strongly suggests that addition of XOs successfully induced a functional xylanolytic system of this species (Fig. 2C).

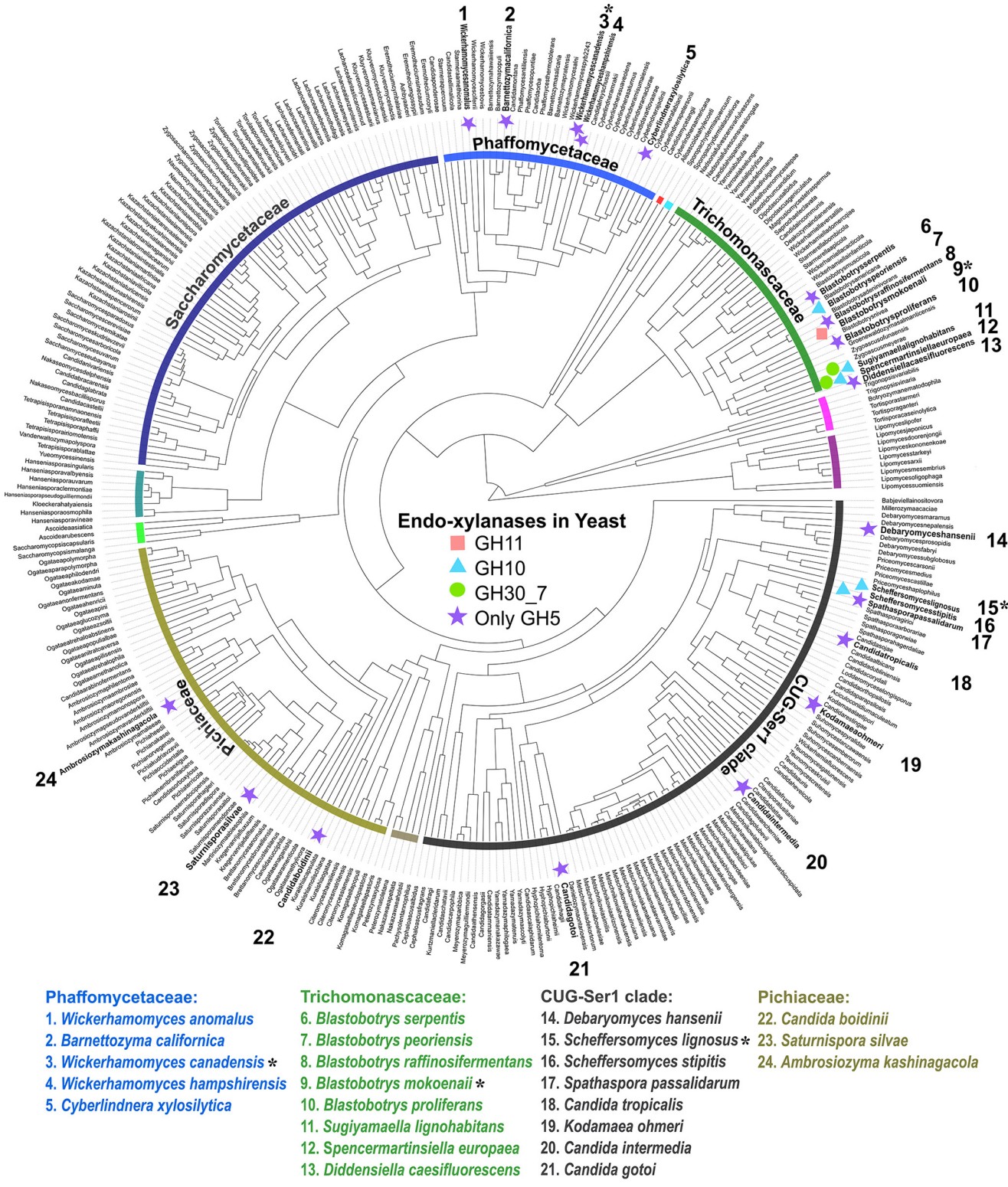

**FIG 1** Overview of predicted endo-xylanases in known xylanolytic yeasts. Putative xylanases are marked next to known xylanolytic yeasts (highlighted in bold and by numbers). Asterisks (*) indicate the three yeasts selected for characterization. GH, glycoside hydrolase.

**Phaffomycetaceae:**
1. *Wickerhamomyces anomalus*
2. *Barnettozyma californica*
3. *Wickerhamomyces canadensis* *
4. *Wickerhamomyces hampshirensis*
5. *Cyberlindnera xylosilytica*

**Trichomonascaceae:**
6. *Blastobotrys serpentis*
7. *Blastobotrys peoriensis*
8. *Blastobotrys raffinosifermentans*
9. *Blastobotrys mokoenaii* *
10. *Blastobotrys proliferans*
11. *Sugiyamaella lignohabitans*
12. *Spencermartinsiella europaea*
13. *Diddensiella caesifluorescens*

**CUG-Ser1 clade:**
14. *Debaryomyces hansenii*
15. *Scheffersomyces lignosus* *
16. *Scheffersomyces stipitis*
17. *Spathaspora passalidarum*
18. *Candida tropicalis*
19. *Kodamaea ohmeri*
20. *Candida intermedia*
21. *Candida gotoi*

**Pichiaceae:**
22. *Candida boidinii*
23. *Saturnispora silvae*
24. *Ambrosiozyma kashinagacola*

Endo-β-1,4-xylanases are the main enzymes responsible for xylan depolymerization, supported by β-xylosidases. The enzymes can be either secreted extracellularly, attached to the cell surface, or intracellular. To determine the subcellular localization of potential xylanase enzymes in the three yeasts, the activities in the secretome, intact live yeast cells

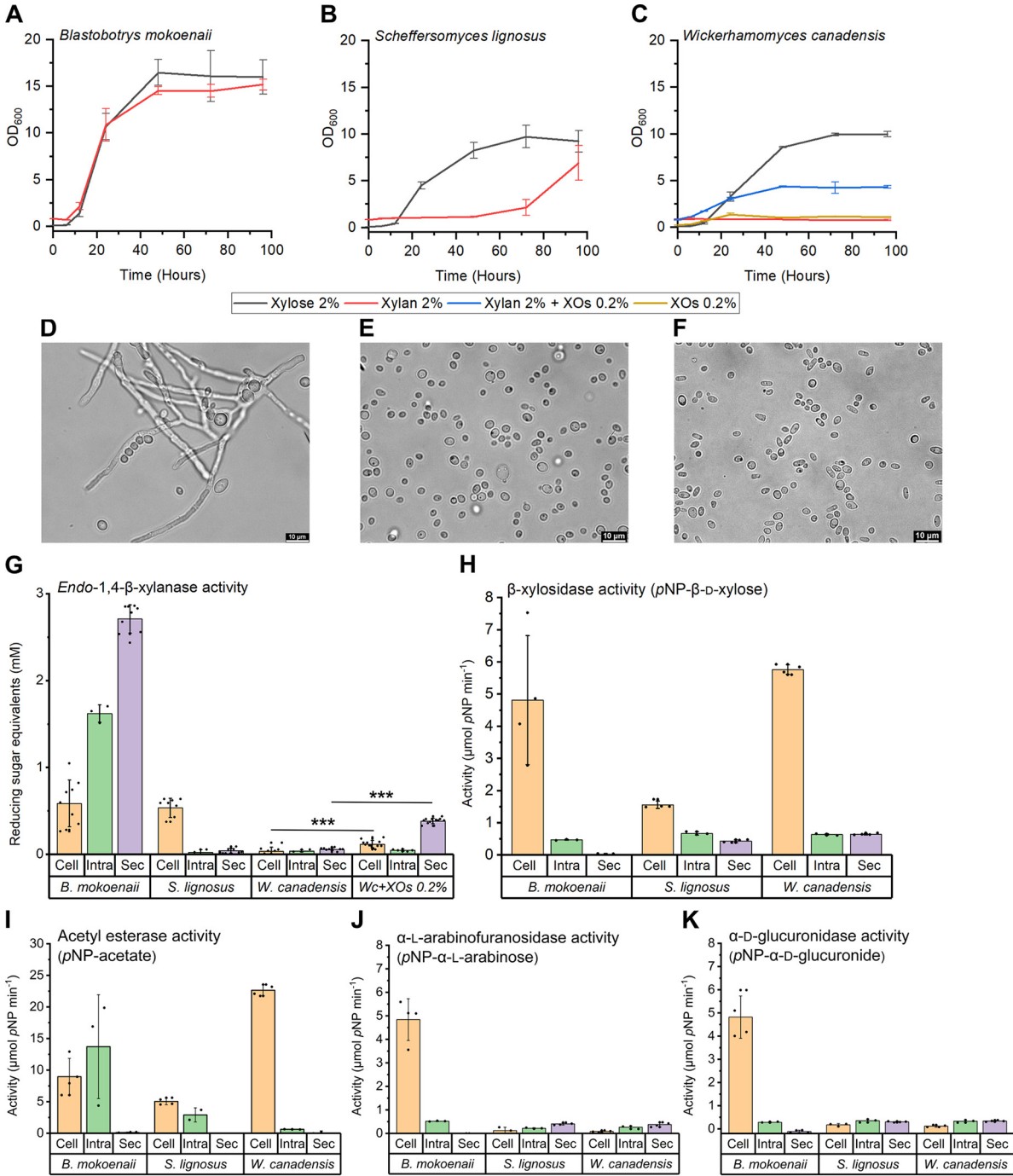

**FIG 2** Yeast growth in beechwood glucuronoxylan and xylanolytic activity localization. (A to C) Yeasts were grown in 10 mL Delft minimal medium with 2% beechwood glucuronoxylan or 2% xylose as the sole carbon source in biological triplicates. XOs, xylooligosaccharides. (D to F) Bright-field microscopy showing yeast morphology from 96-h xylan cultures. (G) Yeast subcellular xylanase activity originating from secretome (Sec), cells (Cell), and intracellular (Intra) from lysed cell fractions were compared using DNS reducing sugar assays in at least triplicate experiments. Values are means ± standard deviations as error bars. Asterisks indicate statistical significance in subcellular activity levels between *W. canadensis* and *W. canadensis* supplemented with XOs. $P$ values of ≤0.0001 (***) were considered significant ($n$ = 12) and evaluated using one-way analysis of variance (ANOVA) with Tukey's test. (H) $\beta$-Xylosidase activity was quantified using $p$-nitrophenyl-$\beta$-D-xylopyranoside; (I) acetyl esterase activity using $p$-nitrophenyl-acetate; (J) $\alpha$-L-arabinofuranosidase activity using $p$-nitrophenyl-$\alpha$-L-arabinofuranoside; and (K) $\alpha$-D-glucuronidase activity using $p$-nitrophenyl-$\alpha$-D-glucuronide.

and lysed cells were determined using the dinitrosalicylic acid (DNS) assay to detect released reducing sugars resulting from xylan cleavage (Fig. 2). Other accessory activities related to xylan deconstruction were monitored using chromogenic substrates to assay for $\beta$-D-xylosidase, $\alpha$-L-arabinofuranosidase, $\alpha$-D-glucuronidase, and acetyl esterase activity.

Clear extracellular xylanase activity was observed only for *B. mokoenaii*, while for *S. lignosus*, xylanolytic activity was cell surface associated. *W. canadensis* cells that had been incubated on xylan alone exhibited no xylanase activity, as expected from the lack of growth, while for cells exposed to XOs together with xylan, extracellular xylanase was detected (significant change from cells incubated on xylan alone; $P < 0.0001$) (Fig. 2G). Accessory activities were comparatively low and mainly cell surface associated in all three yeasts (Fig. 2). All species exhibited $\beta$-xylosidase and esterase activity, though the latter is difficult to pinpoint specifically to xylan degradation. Only *B. mokoenaii* showed significant cell-associated $\alpha$-L-arabinofuranosidase and $\alpha$-glucuronidase activity (Fig. 2J-K), which correlates well with the predicted CAZyme profiles for the three yeasts, where only *B. mokoenaii* possesses putative $\alpha$-L-arabinofuranosidases (GH43, GH51, GH62) and an $\alpha$-D-glucuronidase (GH67). *S. lignosus* possesses a putative $\alpha$-D-glucuronidase (GH115), while *W. canadensis* appears to lack these enzymatic activities altogether (10). Overall, the results show that all three yeasts can degrade xylan polymers and use the released sugars for growth, likely through the combined action of endo-$\beta$-1,4-xylanases, $\beta$-xylosidases, and possibly other accessory enzymes. However, as evident from the differences in growth profiles and detected activities (Fig. 2), the three species use different xylanolytic strategies for xylan degradation.

**Biological roles of the different yeast species.** The fact that the addition of XOs boosts *W. canadensis* growth on xylan indicates that, in nature, this species relies on other microorganisms to initiate the hydrolysis of intact xylan polymers. Secreted enzymes such as the GH11 xylanase from *B. mokoenaii* produce XOs in the extracellular environment, which could also benefit neighboring cells from other species and enable cross-feeding behavior. We co-cultured *W. canadensis* and *B. mokoenaii*, which interestingly, led to synergistic growth behavior, with significantly higher ($P < 0.05$) OD values during the exponential phase (from 24 to 72 h) compared to the *B. mokoenaii* monoculture (Fig. 3A). Moreover, we utilized different fluorescent staining patterns (see Fig. S1 in the supplemental material for controls) of the two species using calcofluor white and FUN-1, to follow their individual growth throughout the co-culture (Fig. 3B to E). *B. mokoenaii* appears to supply *W. canadensis* with XOs by its secreted endo-xylanase, which could also be observed on agar plates, where *B. mokoenaii* created a clearing zone from xylan hydrolysis that upon reaching *W. canadensis* enabled growth (Fig. 3F and G and Videos S1 and S2 for a 21-day time lapse). These results correlate well with the *W. canadensis* growth being induced by the addition of XOs as described above and strongly indicate that externally supplied xylanase activity triggers *W. canadensis* growth on xylan and expression of its own xylanases.

**Biochemical characterization of putative endo-1,4-$\beta$-xylanases.** As all three species showed the ability to metabolize xylan, we sought to characterize the likely responsible xylanases that would initiate its depolymerization. The selection of putative xylanases was based on CAZyme profile predictions, secretome in-gel proteomics from xylan cultures (10), and preliminary flow cell proteomics targeting cell-associated proteins in beechwood GX cultures (our unpublished results). The candidates were from the major xylanase-containing families GH11 (*Bm*Xyn11A) and GH30 (*Bm*GH30_7) in *B. mokoenaii* and GH10 (*Sl*Xyn10A) in *S. lignosus*. As mentioned, *W. canadensis* lacks obvious xylanase candidates but possesses enzymes from GH5, a large polyspecific family with many different $\beta$-1,4-cleaving enzymes (including xylanases), which are further grouped into subfamilies (3, 13). *W. canadensis* encodes enzymes from subfamilies 9 (GH5_9), 12, 22, and 49. Of these, only GH5_22 has been shown to include xylanolytic enzymes, though no reported activity of GH5_49 members exists. Thus, the GH5_22 (*Wc*Xyn5_22A) and GH5_49 (*Wc*Xyn5_49A) enzymes were selected as potential xylanases from this species. A list of all target candidate genes for recombinant expression can be viewed in Table 1. Codon-optimized genes were synthesized, and all targets except *Bm*GH30_7 were successfully expressed in *Pichia pastoris* and purified (gene sequences in List S1 and purified recombinant enzymes in Fig. S2). Each enzyme was assayed at appropriate dilutions (Fig. S3), temperatures (40 to 55°C), and pH (5 to 6) (Fig. S4) corresponding to the growth preferences of each species.

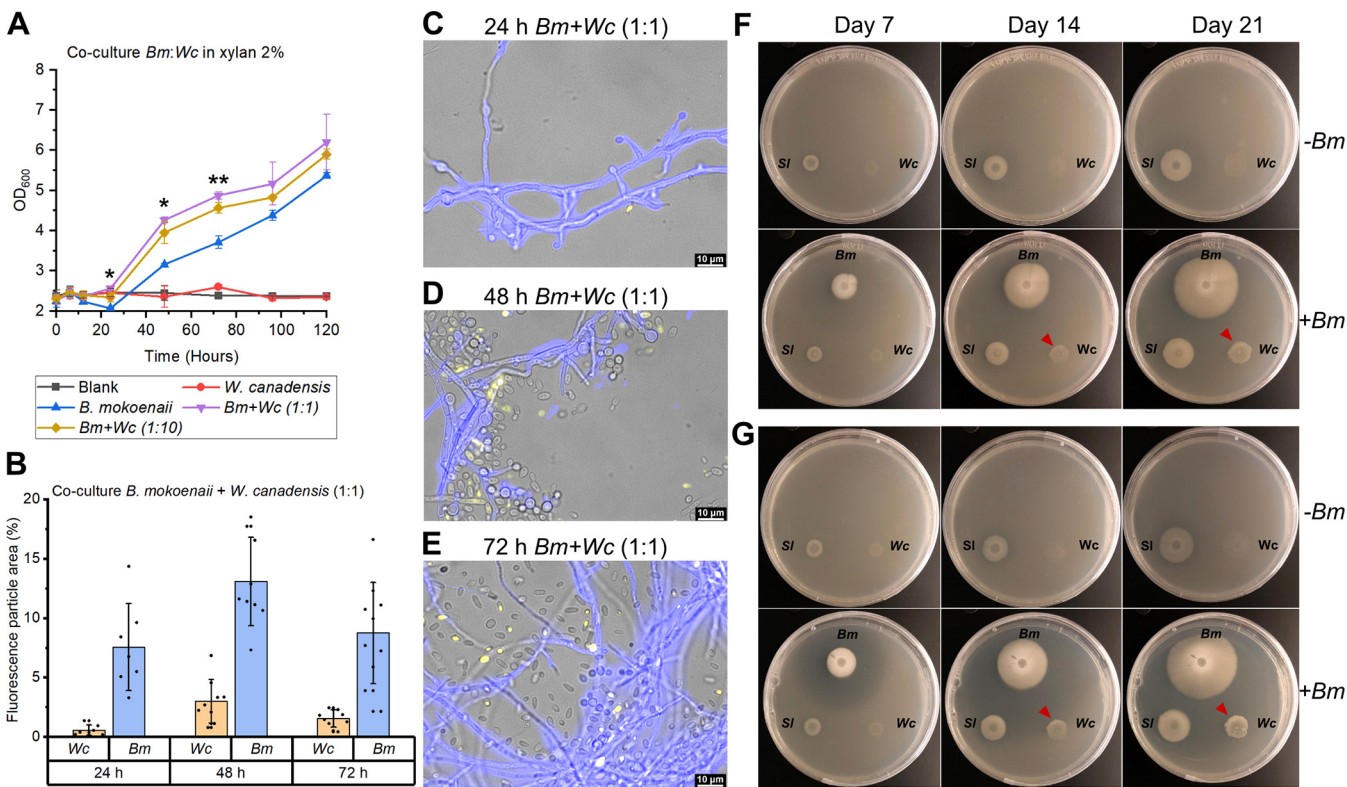

**FIG 3** Co-cultures of yeast growth in xylan. (A) Co-cultures of *B. mokoenaii* and *W. canadensis* at different starting ratios in 2% beechwood glucuronoxylan in triplicates. Values are means ± standard deviations as error bars. Asterisks indicate statistical significance in OD levels at different time points between *B. mokoenaii* mono- and co-culture treatments with *W. canadensis*. P values of ≤0.05 (*) and ≤0.01 (**) were considered significant (*n* = 3) and evaluated using one-way ANOVA with Tukey's test. (B) Distribution in fluorescence particle area (%) of co-cultures of *B. mokoenaii* (blue) and *W. canadensis* (yellow) at an initial ratio of 1:1. (C to E) Representative images from fluorescence microscopy of *B. mokoenaii* and *W. canadensis* (1:1 ratio) (C) after 24 h, (D) after 48 h, and (E) after 72 h. (F and G) Co-cultures on agar plates with (F) 0.4% beechwood glucuronoxylan and (G) 0.4% wheat arabinoxylan over time with (+*Bm*) or without (−*Bm*) the addition of *B. mokoenaii*. The clearing zones correlate with xylanase-mediated xylooligosaccharide release and enable *W. canadensis* growth (red arrows). *Bm*, *Blastobotrys mokoenaii*; *Sl*, *Scheffersomyces lignosus*; *Wc*, *Wickerhamomyces canadensis*.

The specific activities of the purified enzymes were determined on various polysaccharides, biomasses, and synthetic substrates; *Bm*Xyn11A, *Sl*Xyn10A, *Wc*Xyn5_49A, and *Wc*Xyn5_22A all showed endo-$\beta$-1,4-xylanase activity on beechwood GX and wheat AX but no or minor activity on other carbohydrates (Table 2). *Bm*Xyn11A had the highest specific activity of the recombinant enzymes on xylan (4,350 U · mg$^{-1}$), nearly 4-fold higher on beechwood GX than on the closely related GH11 XynB from *Aspergillus niger An*76 (1,146 U · mg$^{-1}$) (14). *Sl*Xyn10A was active on both beechwood GX and wheat AX, though with relatively low activity (13 to 15 U mg$^{-1}$) compared to many bacterial and fungal xylanases (15). *Wc*Xyn5_49A similarly had moderate activity (Table 2), though no direct comparisons within the subfamily can be made, as to the best of our knowledge, it is the first member to be biochemically characterized. While *Wc*Xyn5_49A was mainly active on xylan, *Wc*Xyn5_22A showed activity on both *p*-nitrophenyl-$\beta$-D-xyloside and xylans, albeit low, indicating both $\beta$-xylosidase activity and endo-$\beta$-1,4-xylanase activity. $\beta$-Xylosidase activity was previously reported for *Pc*GH5_22 from the fungus *Phanerochaete*

**TABLE 1** Candidate yeast xylanases expressed in *Pichia pastoris*

| Species | Enzyme family[a] | Codon-optimized protein |
|---|---|---|
| *Blastobotrys mokoenaii*, CBS 8435 | GH11 | *Bm*Xyn11A |
| | GH30_7 | *Bm*GH30_7[b] |
| *Scheffersomyces lignosus*, CBS 4705 | GH10 | *Sl*Xyn10A |
| *Wickerhamomyces canadensis*, CBS 1992 | GH5_49 | *Wc*Xyn5_49A |
| | GH5_22 | *Wc*Xyn5_22A |

[a]GH, glycoside hydrolase.
[b]No expression.

**TABLE 2** Specific activity of recombinant yeast enzymes

| Substrate | Sp act (U · mg$^{-1}$) | | | |
|---|---|---|---|---|
| | *Bm*Xyn11A | *Sl*Xyn10A | *Wc*Xyn5_49A | *Wc*Xyn5_22A |
| Barley $\beta$-glucan 0.5 % | | <0.4 | <0.3 | <0.9 |
| Beechwood glucuronoxylan 1% | 4,350 ± 152 | 14.8 ± 1.3 | 13 ± 2 | <0.8 |
| Galactomannan 1% | | | <0.1 | |
| Laminarin 1% | | <0.6 | <0.6 | |
| Xyloglucan 0.5% | | | <0.3 | <1 |
| Wheat arabinoxylan 1% | 773 ± 39.9 | 12.9 ± 0.91 | 4.3 ± 1 | 1.2 ± 0.2 |
| Birchwood 1% | <1.0 | | <0.4 | 1.5 ± 0.8 |
| Spruce wood 1% | 14.7 ± 1.89 | | | |
| Spruce bark 1% | | <0.8 | | |
| Wheat straw 1% | 7.4 ± 2.10 | <0.7 | <0.3 | 1.6 ± 0.3 |
| *p*-Nitrophenyl-$\beta$-D-xylopyranoside | | | | 0.3 |

*chrysosporium* (16). Although we cannot rule out that the studied yeasts possess additional endo-xylanases, likely including *Bm*GH30_7 and a second GH10 copy in *S. lignosus* and possibly other yet undiscovered enzymes, these results clearly demonstrate that *Bm*Xyn11A, *Sl*Xyn10A, *Wc*Xyn5_22A, and *Wc*Xyn5_49A all are active on GX and AX.

Specific activities (U · mg$^{-1}$) of the recombinant enzymes *Bm*Xyn11A, *Sl*Xyn10A, *Wc*Xyn5_22A, and *Wc*Xyn5_49A were assessed using a DNS reducing sugar assay (except for *p*-nitrophenyl [*p*NP] substrates) using appropriate dilutions (0.002 $\mu$g, 2.2 $\mu$g, 1 $\mu$g, or 2.5 $\mu$g purified protein, respectively). Biomass substrates were milled (particle size, <1 mm). Standard deviations (±) are from triplicate experiments. Activity values of >1 U · mg$^{-1}$ are reported. No activity was observed on carboxymethylcellulose (CMC, 1%), curdlan (1%), glucomannan (1%), *p*-nitrophenyl-acetate, *p*-nitrophenyl-$\alpha$-L-arabinofuranoside or *p*-nitrophenyl-$\alpha$-D-glucuronide.

**Xylan deconstruction and xylooligosaccharide profiles of recombinant xylanases.** To compare and further characterize the identified yeast xylanases, degradation of beechwood GX and wheat AX and their hydrolysis products formed over time were determined using an enzyme concentration of 0.1 $\mu$M. After 24 h, XO profiles were analyzed by high-performance anion-exchange chromatography coupled with pulsed amperometric detection (HPAEC-PAD; Fig. 4A and B). *Bm*Xyn11A reached near-maximum xylan hydrolysis levels after only 15 min (Fig. 4A), releasing mainly smaller XOs (xylose, X1; xylobiose, X2; and xylotriose, X3) (Fig. 4C). *Sl*Xyn10A took longer (8 h) to reach near-maximum hydrolysis levels of both xylans and yielded mainly X2 and X3 from AX and also longer XOs from GX (similar levels of X2-xylohexaose, X6). The differences in XO distributions between *Bm*Xyn11A and *Sl*Xyn10A from GX correspond to previous studies, where GH11 xylanases require three unsubstituted xylose units for attack on the xylan backbone, while GH10 xylanases can accommodate more branched xylans (17, 18). *Wc*Xyn5_49A generated mainly X2 and X3 from both xylans but also xylotetraose (X4) from GX, while *Wc*Xyn5_22A generated minor amounts of XOs, indicating that it is likely not the main xylanase of *W. canadensis* (Fig. 4B). Around X3, overlapping peaks were observed, especially from GX hydrolysis (Fig. S5), which likely represent branched oligosaccharides for which we did not have corresponding standards, but these were tentatively quantified using the X3 standard (Fig. 4). Larger XOs (>X6), many of which are likely branched, were also observed later in the HPAEC-PAD chromatograms (Fig. S5).

To provide further experimental support for the endo-xylanase activity of *Wc*Xyn5_49A, we supplemented *W. canadensis* xylan cultures with different amounts of recombinant *Wc*Xyn5_49 and *Bm*Xyn11A (positive control). Addition of either of the recombinant xylanases resulted in an initial drop followed by a significant increase in OD$_{600}$, indicative of xylan solubilization and cellular growth, respectively (Fig. 4D). Although addition of *Bm*Xyn11A resulted in higher growth rates and final OD$_{600}$ compared to *Wc*Xyn5_49A, the results firmly establish *Wc*Xyn5_49A as an endo-xylanase that can successfully generate XOs and trigger *W. canadensis* growth on xylan.

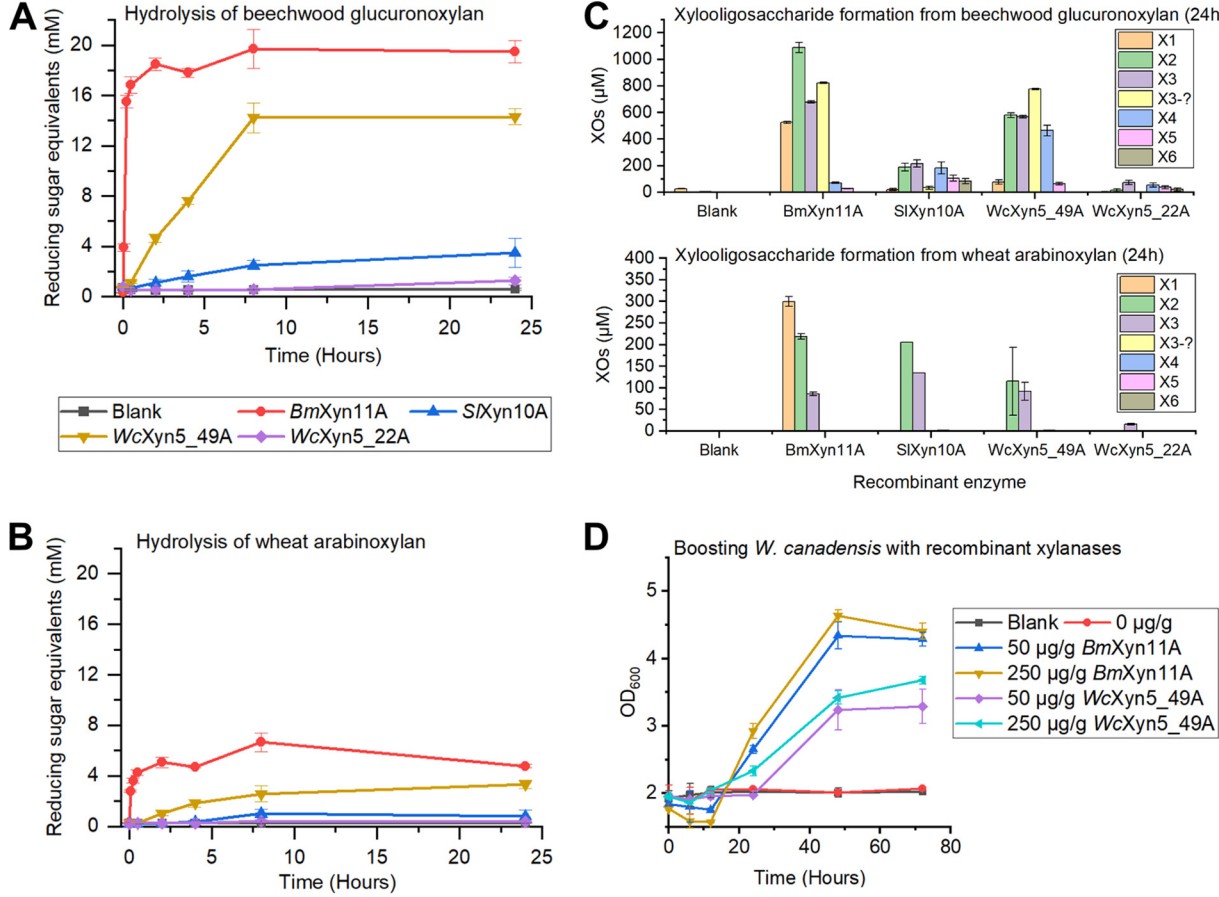

**FIG 4** Effect of xylanase treatment on beechwood glucuronoxylan and wheat arabinoxylan and boosting of yeast growth. (A and B) Hydrolysis of (A) beechwood glucuronoxylan and (B) wheat arabinoxylan by recombinant enzymes over 24 h using 0.1 $\mu$M enzyme in triplicates. (C) Xylooligosaccharide formation from beechwood glucuronoxylan and wheat arabinoxylan after 24 h in duplicates. X1, Xylose; X2, xylobiose; X3, xylotriose; X4, xylotetraose; X5, xylopentaose; X6, xylohexaose; XOs, xylooligosaccharide. (D) Boosting of *W. canadensis* growth in 2% beechwood glucuronoxylan by supplying the cultures with *Bm*Xyn11A and *Wc*Xyn5_49A in lower (50 $\mu$g/g xylan) and higher (250 $\mu$g/g xylan) concentrations in biological triplicates.

**Xylanase structural analyses.** We aimed to solve the structures of the recombinantly expressed xylanases for comparisons with previously solved xylanase structures. We successfully crystallized and solved the structure of *Bm*Xyn11A extending to 1.55-Å resolution, by molecular replacement using an AlphaFold2-predicted structure of *Bm*Xyn11A as the search model. The data have been deposited in the PDB under accession code 8B8E, and collection and refinement statistics are shown in Table S1. The structure of the *P. pastoris*-expressed *Bm*Xyn11A is the first GH11 structure from a budding yeast. The overall structure adopted the expected GH11 $\beta$-jelly roll fold (Fig. 5A), and the asymmetric unit contained five protein molecules (Fig. S6). By superposition of *Bm*Xyn11A with the structure of an *Aspergillus niger* GH11 xylanase in complex with X5 (PDB: 2QZ2; 19), the *Bm*Xyn11A catalytic residues were identified as Glu113 (nucleophile) and Glu204 (proton donor) (Fig. 5B and C). Comparisons with previously solved GH11 members from bacteria and fungi show high conservation, even among kingdoms (Fig. S7), with C$\alpha$-root mean square deviation (RMSD) values of <0.6 Å to *Bm*Xyn11A, which could suggest either retention of this enzyme in *B. mokoenaii* after the split between yeast and filamentous fungi or possibly a recent horizontal gene transfer event.

While we were unable to experimentally solve the structures of the other yeast xylanases, we could predict their structures using AlphaFold2 (20) (Fig. 6). *Sl*Xyn10A conformed to other GH10 enzymes (Fig. 6A), with a C$\alpha$-RMSD value of 0.937 to *Cb*Xyn10C from *Caldicellulosiruptor bescii* (PDB: 5OFK), which enabled identification of the catalytic

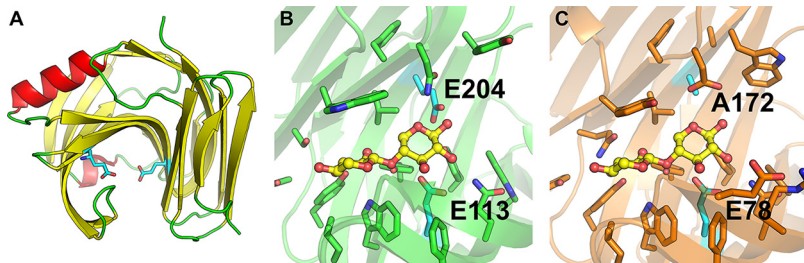

**FIG 5** Crystal structure of *Bm*Xyn11A and comparison with other GH11 members. (A) The overall fold of *Bm*Xyn11A (chain A), with β-sheets in yellow, α-helices in red, loops in green, and catalytic Glu residues as cyan sticks. (B) Closeup of the active site cleft of *Bm*Xyn11A (green) superimposed with the bound oligosaccharide in yellow ball-and-stick representation derived from the *Aspergillus niger* GH11 xylanase (PDB: 2QZ2, inactivated E170A variant, crystallized with X5); (C) shown in the same orientation.

residues of *Sl*Xyn10A: E163 (proton donor) and E270 (nucleophile) (21). The active site appears to be an extended cleft, which suggests an ability to accommodate long xylan chains or XOs and, likely, substitutions depending on the angle of the backbone xylose moieties (Fig. 6A). The active sites of *Sl*Xyn10A and *Cb*Xyn10C are nearly identical in each sugar-binding subsite and infer six or more subsites in *Sl*Xyn10A (21). *Wc*Xyn5_22A and *Wc*Xyn5_49A both adopted the expected GH5 $(\alpha/\beta)_8$ barrel, though *Wc*Xyn5_49A also was appended with a C-terminal bundle of four alpha helices (Fig. 6B, panel i, green; residues 400 to end) of unknown function; it did not show any similarity to known carbohydrate binding module (CBM) families (3, 22). Such helical bundle inserts have been

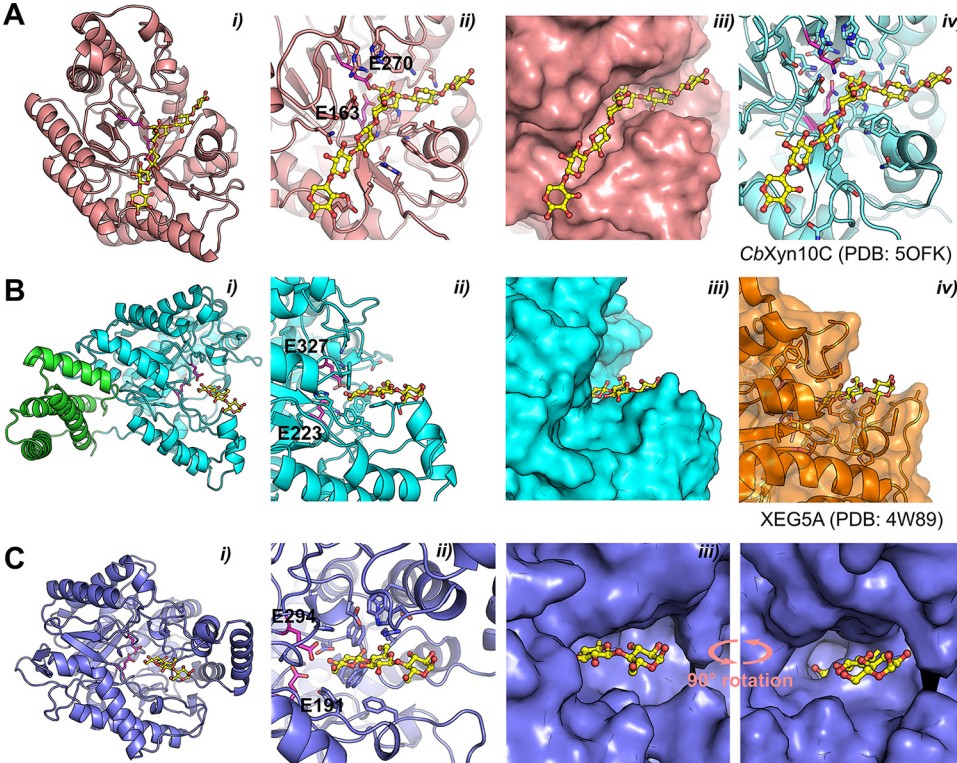

**FIG 6** Predicted AlphaFold 2 structures for GH5 and GH10 xylanases and β-xylosidases characterized in this study. (A) *Sl*Xyn10A. (B) *Wc*Xyn5_49A. (C) *Wc*Xyn5_22A. (i) Overview of each enzyme. The helical bundle in *Wc*Xyn5_49A is shown as green sticks. The catalytic Glu residues are magenta sticks. In the GH5 structures, a superimposed oligonucleotide from the structure of XEG5A (PDB: 4W89) is shown as yellow sticks; in the GH10 structure the oligonucleotide is superimposed from *Cb*Xyn10C (PDB: 5OFK). (ii) Closeups of each active site. (iii) Surface representation, highlighting the different shapes of the binding clefts. In panel Ciii, a 90° rotated view reveals the deep cleft of *Wc*Xyn5_22A with limited surface accessibility. (iv) Reference structures of *Cb*Xyn10C and XEG5A. Figure S7 visualizes the pLDDT scores for each predicted model.

shown to confer thermal stability in other proteins (23, 24), but this does not appear to be the case for *Wc*Xyn5_49A, as it did not exhibit significant thermal stability (Fig. S4). By comparison of the GH5 enzymes to XEG5A (PDB: 4W89 [25]), the catalytic residues were identified for *Wc*Xyn5_49A as Glu223 (proton donor) and Glu327 (nucleophile) (Fig. 6B, panel ii) and for *Wc*Xyn5_22A as Glu191 (proton donor) and Glu294 (nucleophile) (Fig. 6C, panel ii). The active sites of *Sl*Xyn10A and *Wc*Xyn5_49A adopted expected clefts able to accommodate extended polysaccharide substrates (Fig. 6A and B, panels iii), but a helical insert outside the active site of *Wc*Xyn5_22A (Fig. 6C, panels i to iii) creates a large ridge that appears to block positive subsites (toward the reducing end). Such an apparent blockage of the active site could suggest exo-activity rather than endo, though this does not correlate with the biochemical analyses (Fig. 4), and the modeled positioning of this insert may not be of biological relevance despite high confidence scores (Fig. S8). In contrast, *Wc*Xyn5_49A possesses a highly accessible active-site cleft, suggesting it can accommodate large substrates (Fig. 6B, panel ii). Future experimental determination or modeling of ligand-bound structures of the yeast xylanases may reveal deeper molecular information about their specificities.

## DISCUSSION

Our previous work using comparative genomics among known xylan-degrading budding yeasts revealed different setups of xylanolytic enzymes, where many species lack xylanases from typical GH families (10). The CAZy family GH5 has over 50 subfamilies that mainly contain enzymes cleaving various $\beta$-(1, 4)-linked oligo- and polysaccharides (3) and therefore stood out as a likely source of endo-$\beta$-1,4-xylanases in the yeasts without obvious candidates (2). In this study, we demonstrate experimentally that three xylanolytic yeasts, *B. mokoenaii*, *S. lignosus*, and *W. canadensis*, isolated from different environments and belonging to different phylogenetic clades, have evolved distinct strategies to degrade xylan that manifests as different growth behaviors on this carbon source. We also provide experimental validation of xylanase activity for the yeast enzymes *Bm*Xyn11A, *Sl*Xyn10A, *Wc*Xyn5_22A, and *Wc*Xyn5_49A, the latter being from a previously uncharacterized CAZyme subfamily.

The GH5_49 subfamily appears enriched in yeast compared to filamentous fungi and bacteria (3). In the data set of 332 genome-sequenced budding yeasts, 319 have predicted GH5_49 genes and 126 possess GH5_22 genes (10), for which there has previously been no (GH5_49) or little (GH5_22) experimental characterization. Although 16 of the 24 known xylanolytic yeasts in the data set contain both GH5_22 and GH5_49 genes, there are also many yeasts that have such subfamily genes but still are not known to depolymerize and grow on xylan (3). It is clear that additional characterization of GH5_22 and GH5_49 members is needed to verify whether they generally act as xylanases or have different functions in other yeast and non-yeast species. Exploring possible synergies with other CAZymes would further shed light on their biological role(s).

In the three species investigated here, other enzymes may also have roles in xylan turnover, but our physiological and biochemical analyses begin to paint a picture of their highly divergent xylanolytic strategies (Fig. 7). *B. mokoenaii* is the standout species thanks to its secreted powerful *Bm*Xyn11A (10), which likely facilitates the rapid growth on xylan. Also, the pseudomycelial growth morphology of *B. mokoenaii* may be an advantage in its native soil environment, similar to many filamentous fungi (26). Yeasts that possess GH11 xylanases seem rare, and we have only been able to find one other ascomycetous yeast, *Aureobasidium pullulans*, and a few basidiomycetous yeasts containing GH11 genes, all of which display superior capacity to degrade xylan (2, 27). In contrast to *B. mokoenaii* and *W. canadensis*, the xylanase activity of *S. lignosus* was mainly localized to the cell surface, which is also a feature of the $\beta$-xylosidase activity of all three species. Possibly, the native environment of *S. lignosus* in the presumably highly competitive insect gut has exerted a selective pressure for more selfish nutrient acquisition behavior, where full secretion of xylanases into the extracellular environment does not promote fitness. Similar selfish behavior of microorganisms has been

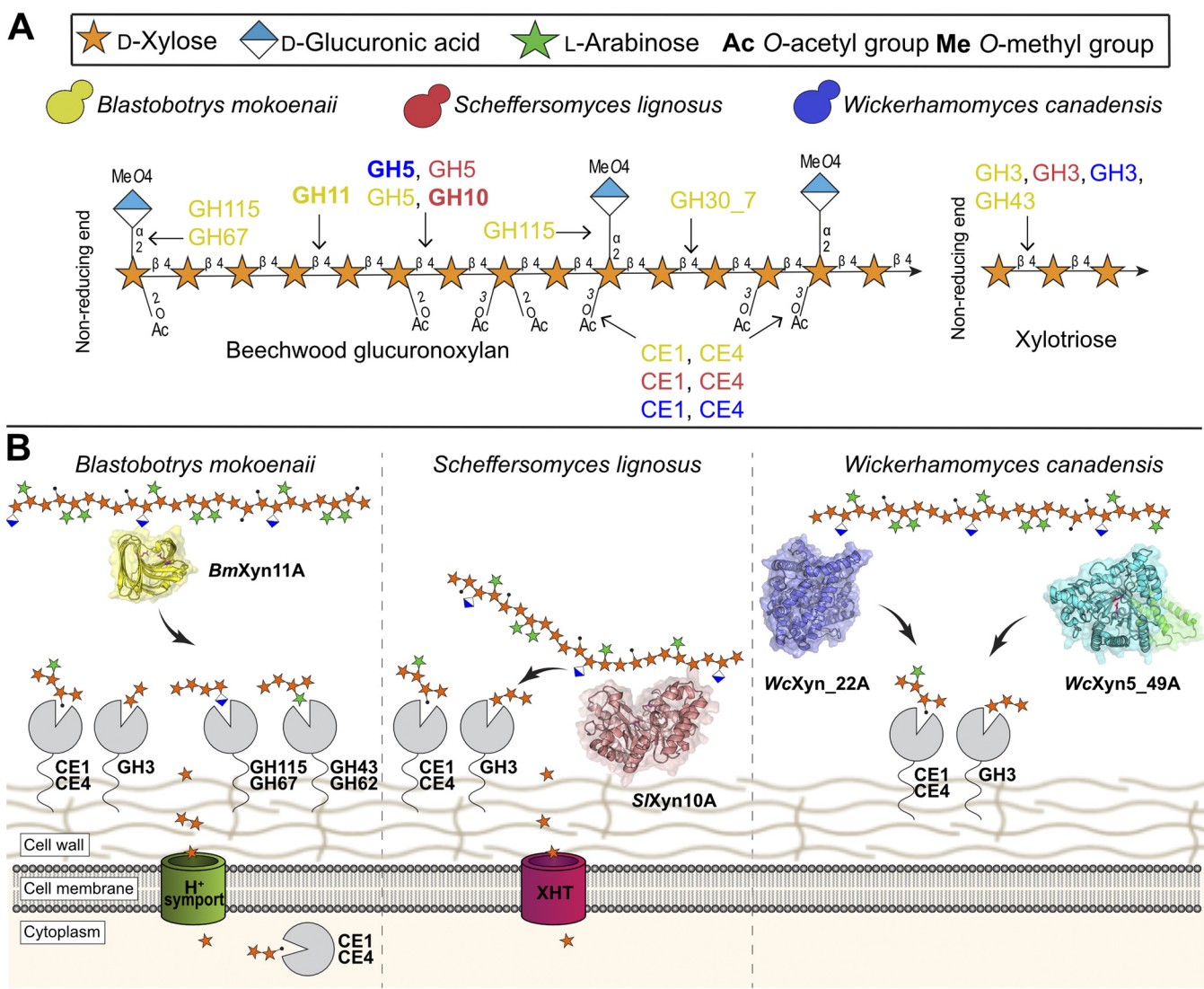

**FIG 7** Schematic view of the xylanolytic strategies of the three investigated species. (A) Predicted xylanolytic CAZymes (10) of *B. mokoenaii*, *S. lignosus*, and *W. canadensis* represented by color (yellow, red, and blue, respectively). Arrows from individual enzyme families indicate enzyme activity in beechwood glucuronoxylan degradation. CE, carbohydrate esterase; GH, glycoside hydrolase. (B) Xylan. *B. mokoenaii* utilizes a secreted xylanase which depolymerizes xylan away from the cells and liberates oligosaccharides in the process. *S. lignosus* instead maintains the xylanase activity close to the cell surface, which might give a competitive advantage in its native environment. *W. canadensis* secretes xylanases similarly to *B. mokoenaii*, but only after sensing xylooligosaccharides released by xylanase action from neighboring cells. Putative enzymes, as indicated by activity on chromogenic model substrates, and therefore remaining speculative, are shown in gray together with assumed CAZy family memberships based on previous work (10). Acetyl groups are shown as black dots. XHT, xylose/glucose transporters.

demonstrated, for example, in the human gut, where certain polysaccharide-metabolizing bacteria tether both enzymes and carbohydrate-binding proteins to the cell surface to limit leakage of smaller sugars (28, 29). Intracellular enzymes require transporters that can internalize the substrate first, which are uncommon for larger oligosaccharides (30). The surface-located $\beta$-xylosidase activity of the three yeasts studied here would, however, suggest that mainly small sugars are transported over the plasma membranes. The detected esterase activity in all yeasts is attributed to CEs from families CE1 or CE4, as these are the only ones found in these species (10). More detailed studies on acetylated xylan or XOs would, however, be needed to conclusively demonstrate acetyl xylan esterase or, potentially, feruloyl esterase activity.

We observed that *W. canadensis*, even when pregrown on xylose, cannot grow on the beechwood GX used in this study. Apparently, it lacks the ability to sense intact xylan polymers, but its xylanolytic machinery was inducible by small amounts of XOs,

or by addition of xylanases that presumably lead to the same products. This phenomenon has also been observed previously for some other xylanolytic yeasts (31). It is possible that *W. canadensis* can sense longer oligosaccharides than tested here, similar to some bacteria which require medium- to large-sized xylan fragments to induce endo-$\beta$-1,4-xylanase expression (32). In its natural habitat, it is thus likely that *W. canadensis* needs neighboring primary degraders of xylan releasing XOs for it to kickstart its own xylanolytic machinery. Biomass-degrading microbes are not presumed to act alone in nature, but to act in concert with other species, by either collaborating or competing for the sugars. In this sense, *W. canadensis* seems to have an opportunistic xylanolytic strategy, which possibly could be advantageous by not expending resources for constitutive xylanase production and instead responding only when xylan is being degraded nearby. It is interesting to note that *W. canadensis* displays relatively high $\beta$-xylosidase activities, which may suggest a competitive advantage for XOs released during active xylan degradation and a successful xylan scavenging behavior.

Our findings on xylanolytic yeasts may provide a starting point for further fundamental research on their biomass-degrading enzymes and cellular physiology. Future characterization of other species will further reveal how commonplace the proposed biological role(s) of the yeasts studied here are in global biomass decay ecosystems. Co-culturing species from the same, or similar, environments would be of high interest to compare with our *B. mokoenaii-W. canadensis* cultures, as such experiments could showcase whether other opportunistic species like *W. canadensis* can exhibit similar behavior also on other types of polysaccharides. The findings in this study are also of industrial importance. Mono- and co-cultures of xylanolytic yeasts can be used in consolidated bioprocesses, where simultaneous hydrolysis of biomass/xylan and conversion into product are carried out by the microorganisms rather than the former being performed by expensive enzyme cocktails (33). The yeasts characterized here might be used as future cell factories in their own right or as donors of genes to already established yeast cell factories based on, e.g., *Saccharomyces cerevisiae*. Moreover, the enzymes studied here may also find applications in food and feed production or the pulp and paper industries. For example, *Bm*Xyn11A showed no activity on CMC, which is a biotechnologically attractive trait in pulp bleaching (34).

In summary, our study shows that among dissimilar yeast species, there are wide-ranging xylan degradation strategies and enzymes. Understanding these strategies may be highly useful to transfer such capabilities to an industrial yeast host such as *S. cerevisiae*, which has a similar yeast morphology and might benefit from cell-attached CAZymes, compared to enzymes from filamentous fungi and bacteria that are typically secreted as soluble enzymes. We conclude that knowledge of novel xylanolytic enzymes and strategies may help scientists to find new strategies for metabolic engineering, cell-factory design, and synthetic biology of yeasts in the future.

## MATERIALS AND METHODS

**Yeast growth characterization in xylan liquid cultures.** Yeasts strains were ordered from the ARS Culture Collection, USA (NRRL; https://nrrl.ncaur.usda.gov/). Yeast growth in beechwood glucuronoxylan or xylose was determined by inoculating yeasts with a starting $OD_{600}$ of 0.1 in 10 mL Delft minimal medium (5 g $L^{-1}$ ammonium sulfate, 3 g $L^{-1}$ potassium phosphate, 1 g $L^{-1}$ magnesium sulfate, vitamins, and trace metals as described previously [35]) additionally containing either 20 g $L^{-1}$ (wt/vol) of beechwood glucuronoxylan (Megazyme, Ireland) or 20 g $L^{-1}$ xylose (wt/vol). Cultures were kept at either 30°C (for *B. mokoenaii* CBS 8435, Y-27120) or room temperature (~21°C; for *S. lignosus* CBS 4705, Y-12856 and *W. canadensis* CBS 1992, Y-1888) at 150 rpm for 96 h in 100-mL baffled flasks. Xylooligosaccharides (<95% XOs) from corncob (Carlroth, Germany) were added at 2 g $L^{-1}$ (wt/vol) to kickstart *W. canadensis* growth, with or without additional xylan. Yeasts were precultured in Delft minimal medium with 20 g $L^{-1}$ xylose and washed in Delft minimal medium without a carbon source before inoculation into xylan or xylose. For co-cultures, yeast precultures (Delft minimal medium with 20 g $L^{-1}$ xylose) were inoculated separately with a starting $OD_{600}$ of 0.1 or in different ratios (1:1 and 1:10) in Delft medium plus 2% beechwood glucuronoxylan (Megazyme, Ireland) for 120 h at room temperature (RT) at 200 rpm. For growth boosting with recombinant enzymes, 50 $\mu$g/g xylan or 250 $\mu$g/g xylan enzymes was supplemented to cultures.

**Yeast subcellular enzyme activities.** To quantify the subcellular endo-$\beta$-1,4-xylanase activity during xylan degradation (secretome, cell-associated or intracellular), cells were harvested by centrifugation

(3,000 × $g$, 15 min) and divided into two fractions. Each cell fraction was washed in Delft minimal medium (without carbon source); one fraction was kept for cell-associated activity, and the other (OD$_{600}$, 5) was lysed by eight cycles of bead beating at 8,000 rpm, 30 s, followed by the addition of Y-PER (yeast protein extraction reagent; Pierce, Rockford, IL, USA) breaking buffer with 25 mM 1,4-dithiothreitol. Lysis efficiency was monitored microscopically. The soluble intracellular fraction was isolated by centrifugation (13,000 × $g$, 5 min) and assayed alongside the cell-free supernatant (secretome) and the intact cell pellets. The assay contained 10 g L$^{-1}$ wheat AX or beechwood GX and 50 mM sodium acetate buffer (pH 5.5) added to 25 $\mu$L of secretome or cell-free intracellular fraction, while cells were assayed at an OD$_{600}$ of 0.25 and mixed in a 96-well plate. The mixture was incubated at 30°C for 30 min, followed by immediate chilling on ice. The 96-well plate was then centrifuged at 4°C (4,000 × $g$, 5 min) to remove intact yeast cells, and 50 $\mu$L was transferred to a new 96-well plate. Reducing sugar ends were determined by the dinitrosalicylic acid (DNS) method (36). All enzymatic measurements were performed in triplicates. For debranching activities, the subcellular cell fractions (using the same volumes as mentioned above and cells assayed at an OD$_{600}$ of 0.25) were incubated with either 2.5 mM $p$-nitrophenyl-acetate, $p$-nitrophenyl-$\alpha$-L-arabinofuranoside, $p$-nitrophenyl-$\alpha$-D-glucuronide or $p$-nitrophenyl-$\beta$-D-xylopyranoside in 200-$\mu$L reactions containing 20 mM sodium phosphate (pH 7). The reactions were incubated at 30°C for 20 min at 350 rpm in 96-well plates, which were then centrifuged at 4°C (4,000 × $g$, 5 min) to remove intact yeast cells, and 100 $\mu$L was transferred to a new 96-well plate, where $p$-nitrophenol was quantified at 405 nm over 30 min.

**Bright-field and fluorescent microscopy of yeast co-cultures.** Yeast cells from xylan mono- and co-cultures were sampled (100 $\mu$L) at different time points (24, 48, 72, and 96 h), centrifuged (10,000 × $g$, 5 min), and washed in 1 mL 10 mM HEPES buffer, pH 7.2. Co-cultures were stained with 0.15 $\mu$L FUN-1 and 0.5 $\mu$L calcofluor white in 100 $\mu$L 10 mM HEPES buffer, pH 7.2, plus 2% glucose using a LIVE/DEAD yeast viability kit (Thermo Fisher, Germany) for 30 min in the dark at RT. Fluorescence images were recorded using a Leica DFC360 FX microscope (Germany) with a 100×oil immersion objective and a DFC 360 FX camera. Excitation filters for yellow fluorescent protein (YFP; 480 to 520 nm) were used for FUN-1 using an exposure time of 75 ms with 1.8 gain, and an excitation filter A4 (320 to 400 nm) was used for calcofluor white using an exposure time of 5 ms and 1.8 gain. Emission spectra were designated yellow and blue for FUN-1 and calcofluor white, respectively, in LAS X (Leica) software. Total fluorescent particle area and pixel intensity values in the 12-bit images (1,392 by 1,040 pixels) were used to differentiate and quantify blue and yellow fluorescence using ImageJ (Fiji) software.

**Agar plate xylan sensing assay.** Washed yeast precultures (10 $\mu$L; Delft minimal medium with 20 g L$^{-1}$ xylose) with a starting OD$_{600}$ of 5 were pipetted onto Delft minimal medium agar plates (2%) containing 0.4% beechwood glucuronoxylan (Megazyme, Ireland) or wheat arabinoxylan (Megazyme) with appropriate distances between strains. Plates were incubated at room temperature for 21 days, and pictures were taken daily to follow xylan clearing zones and yeast colony growth.

**Cloning and protein production.** Genes encoding the putative proteins $Bm$Xyn11A, $Bm$GH30_7, $Sl$Xyn10A, $Sl$GH5_22, $Wc$Xyn5_22A, and $Wc$Xyn5_49A (sequence IDs in List S1) were codon optimized and synthesized by Twist Bioscience (USA) for heterologous expression in $P.\ pastoris$ X-33 and cloned into pPICZ$\alpha$ A vectors using EasySelect (Thermo Fisher, Germany). Native signal peptides were removed from $Bm$Xyn11A and $Sl$Xyn10A. The vector was digested with EcoRI and SalI, and inserts were ligated by T4 ligase. Recombinant pPICZ$\alpha$ A constructs contained yeast $\alpha$-secretion factor, candidate gene, and His$_6$ tag and were transformed into $E.\ coli$ DH5$\alpha$ One Shot Top10 cells (Invitrogen). Transformants were selected based on Zeocin resistance (25 $\mu$g mL$^{-1}$) confirmed by colony PCR. Vector propagation was performed in 1.5-mL LB (low salt plus Zeocin, 75 $\mu$g mL$^{-1}$) using a GeneJET PCR purification kit (Thermo Fisher). $P.\ pastoris$ cells were transformed using ~50 $\mu$g linearized pPICZ$\alpha$ A by electroporation, and clones were selected using Zeocin (150 $\mu$g mL$^{-1}$) on yeast extract-peptone-dextrose (YPD) plates with 1 M sorbitol, while colony PCR confirmed the presence of recombinant vectors. $P.\ pastoris$ clones were grown in small scale (5 mL) to confirm recombinant enzyme activity before scale-up to 1-L baffled flasks at 28°C in 400 mL rich buffered glycerol-complex medium (BMGY). Methanol (1%, vol/vol) was used to induce expression in buffered methanol-complex medium (BMMY) as described in the EasySelect Pichia Expression kit, and the cells were grown for 6 days as previously described (37).

**Enzyme purification.** Proteins were purified by immobilized metal affinity chromatography (IMAC) using Ni-Sepharose excel resin (GE Healthcare, USA). The column was first washed with 5 column volumes of loading buffer (50 mM Tris [TRIS], pH 8, 250 mM NaCl). After loading and washing of bound proteins, they were eluted using the same buffer with an additional 250 mM imidazole. $Sl$Xyn10A was deglycosylated with endo H (NEB, USA) according to the supplier's protocol. $Wc$Xyn5_22A and $Wc$Xyn5_49A were further purified using anion-exchange chromatography on a HiTrap SP HP column (Cytiva) using 50 mM TRIS (pH 8) as the loading buffer and elution using a linear gradient with 0 to 1 M NaCl. Protein purity was evaluated by SDS-PAGE. A NanoDrop 2000 spectrophotometer (Thermo Fisher Scientific, Germany) was used to determine protein concentration using the predicted values for molecular weights and extinction coefficients (Expasy ProtParam server).

**Biochemical characterization.** Endo-$\beta$-1,4-xylanase activity was assayed using a 200-$\mu$L mixture of 10 g L$^{-1}$ beechwood GX (Megazyme, Ireland) in 50 mM sodium acetate buffer (pH 5.5) using 10 $\mu$L purified enzyme at suitable dilutions (Fig. S3). The mixture was incubated for 30 min, followed by immediate chilling on ice and inactivation at 98°C for 5 min. For time course measurements, 50- to 100-$\mu$L aliquots were sampled at 0, 0.5, 2, 4, 8, and 24 h and immediately heat-inactivated and stored at −20°C, before reducing sugar ends were determined using the DNS method. For pH and temperature optimum measurements, the following buffers were used at 100 mM: sodium acetate pH 5 or pH 5.5, sodium phosphate pH 6 to 8, sodium citrate pH 3 to 4, 2-(N-morpholino)ethanesulfonic acid (MES) pH 5 to 6, and

*N*-cyclohexyl-2-aminoethanesulfonic acid (CHES) pH 8 to 10. In other polysaccharides, including wheat arabinoxylan (Megazyme), birchwood glucuronoxylan (Sigma-Aldrich, Germany), xyloglucan (tamarind, Megazyme), mixed-linkage $\beta$-1,3/1,4-glucan (barley, Megazyme), galactomannan (guar/locust bean gum, Sigma-Aldrich), glucomannan (konjac, Sigma-Aldrich), curdlan (Merck, USA), pectin (citrus, Sigma-Aldrich), laminarin (*Laminaria digitata*, Sigma-Aldrich), carboxymethyl cellulose (Sigma-Aldrich), and the ball-milled ($\sim$40 $\mu$m) local Swedish biomasses (spruce wood, birchwood, wheat straw, and spruce bark), reducing sugar ends released by purified recombinant enzymes were determined using the DNS method. Specific activity on *p*NP-acetate, *p*NP-$\beta$-xyloside, *p*NP-$\alpha$-L-arabinofuranoside and *p*NP-$\alpha$-glucuronide (2.5 mM), measured using 0.002 $\mu$g (*Bm*Xyn11A), 2.2 $\mu$g (*Sl*Xyn10A), 1 $\mu$g (*Wc*Xyn5_22A), or 2.5 $\mu$g (*Wc*Xyn5_49A) purified protein, was determined in 20-min reactions as described earlier.

**Xylooligosaccharide analysis by ion chromatography.** XO formation from hydrolysis of beechwood glucuronoxylan (Megazyme, Ireland) and wheat arabinoxylan (Megazyme) by recombinant enzymes was analyzed after 24-h reactions (0.1 $\mu$M enzyme, using 1% xylan, at 40°C in 1,000 $\mu$L total volume). Then, 100-$\mu$L aliquots were inactivated by boiling, diluted in water, and centrifuged (5 min, 13,000 rpm), and the supernatants were filtered (0.22 $\mu$m) before storage at 4°C until analysis by high-performance anion-exchange chromatography coupled with pulsed amperometric detection (HPAEC-PAD) using an ICS-5000 system (Dionex, USA) at 25°C. Separation of hydrolysis products was performed with a CarboPac PA200 (250 mm by 3 mm) column (Thermo Fisher, Germany) and the following eluents using a flow rate of 0.5 mL min$^{-1}$: A, water; B, 300 mM sodium hydroxide; and C, 100 mM sodium hydroxide and 1 M sodium acetate. The mobile-phase gradient can be found in Table S2. Standards (5 to 1,000 $\mu$M) of xylose (X1), xylobiose (X2), xylotriose (X3), xylotetraose (X4) xylopentaose (X5), and xylohexaose (X6) (Megazyme) were used for quantitation.

**Protein crystallization, structure determination, and molecular modeling.** Prior to crystallization, a sample of *Bm*Xyn11A was deglycosylated using endo H (RT, overnight). This sample and the *P. pastoris*-expressed untreated enzyme were further purified using size exclusion chromatography, using a HiLoad 16/600 Superdex 200-pg size exclusion column (Cytiva), operated by an ÄKTA Explorer fast protein liquid chromatography (FPLC) instrument using 25 mM Tris, pH 8.0, as the buffer. Comparison with a calibration curve of protein standards confirmed that the enzyme was monomeric in both cases and showed a small degree of glycosylation in the case of the untreated enzyme. The enzymes were concentrated using Amicon 10- kDa-cutoff centrifugal filter units to 65 mg mL$^{-1}$ for the untreated *Bm*Xyn11A and 12 mg mL$^{-1}$ for the endo H treatment. Both proteins were subjected to crystallization screening using a Mosquito robot in sitting-drop vapor diffusion trays at 20°C, with drops of 0.3 $\mu$L protein plus 0.3 $\mu$L reservoir solution, 40 mL reservoir volume, using the JCSG+, Morpheus, and PACT Premier screening kits (Molecular Dimensions). Crystals were not obtained for the endo H-treated enzyme. For the untreated *Bm*Xyn11A, rod-shaped crystals were obtained in 2 weeks from condition D12 from the JCSG+ kit (0.04 M potassium phosphate monobasic, 16% [wt/vol] polyethylene glycol [PEG] 8000, 20% [wt/vol] glycerol, 65 mg/mL *Bm*Xyn11A). Optimized, large crystals were obtained in 3 weeks under the same conditions with 32.5 mg/mL *Bm*Xyn11A. These were flash-frozen in liquid nitrogen. Diffraction data were obtained on 22 April 2022 on the BioMAX beamline at MAX IV (Lund, Sweden) at 100 K. The data set was collected at 0.1° increments over 360° and processed using Mosflm (38), and the structure was solved by molecular replacement with Phaser in the Phenix program suite (39), using an AlphaFold 2 model of *Bm*Xyn11A as the search model (20, 40). The solvent content indicated five molecules in the asymmetric unit. The protein structure was manually rebuilt after phasing using Coot (41) and refined using Phenix Refine. $R_{free}$ was monitored using 5% of the diffraction data selected at random prior to refinement. Detailed data collection and refinement statistics are outlined in Table S1. The coordinates for the structure were validated and deposited in the PDB with accession code 8B8E. Structure prediction models of *Wc*Xyn5_22A and *Wc*Xyn5_49A were produced using AlphaFold 2 software (20, 40).

## SUPPLEMENTAL MATERIAL

Supplemental material is available online only.
**SUPPLEMENTAL FILE 1**, MP4 file, 0.3 MB.
**SUPPLEMENTAL FILE 2**, MP4 file, 0.3 MB.
**SUPPLEMENTAL FILE 3**, DOCX file, 4.6 MB.

## ACKNOWLEDGMENTS

We thank the ARS Culture Collection for providing yeast cultures and Gisela Brändén from the University of Gothenburg for her assistance in protein X-ray crystallography. We acknowledge the MAX IV Laboratory for time on the beamline BioMAX under proposal 20200093.

Research conducted at MAX IV, a Swedish national user facility, is supported by the Swedish Research council under contract 2018-07152, the Swedish Governmental Agency for Innovation Systems under contract 2018-04969, and Formas under contract 2019-02496. The AlphaFold 2 structure predictions were enabled by resources provided by the Swedish National Infrastructure for Computing (SNIC) at Chalmers Centre for

Computational Science and Engineering, partially funded by the Swedish Research Council through grant agreement 2018-05973.

C.G., J.L., and J.L.R. conceived the project; C.G., J.L., and J.L.R. designed the experiments; J.L.R., A.S.R., and T.C. performed the experiments. J.L.R. performed growth cultures, enzymatic assays, and recombinant protein expression. A.S.R. and J.L.R. performed protein purification and characterization. T.C. performed crystallography and AlphaFold predictions together with J.L. J.L.R., A.S.R., and T.C. interpreted the data and wrote the manuscript. J.L.R., C.G., J.L., A.S.R., and T.C. revised the manuscript. All authors read and approved the final manuscript.

This work has received funding from the European Union's Horizon 2020 Research and Innovation Framework Program under grant agreement 964430.

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
