## [Reviewer comments · Microbiology Spectrum]

Microbiology Spectrum

Yeasts have evolved divergent enzyme strategies to deconstruct and metabolize xylan

Jonas Ravn, Amanda Sørensen Ristinmaa, Tom Coleman, Johan Larsbrink, and Cecilia Geijer

Corresponding Author(s): Johan Larsbrink and Cecilia Geijer, Chalmers tekniska högskola AB

Review Timeline:

Submission Date:	January 15, 2023
Editorial Decision:	March 22, 2023
Revision Received:	April 6, 2023
Accepted:	April 8, 2023

Editor: Florian Freimoser

Reviewer(s): Disclosure of reviewer identity is with reference to reviewer comments included in decision letter(s). The following individuals involved in review of your submission have agreed to reveal their identity: Greg Potter (Reviewer #1)

Transaction Report:

DOI: <https://doi.org/10.1128/spectrum.00245-23>

March 22, 2023

Dr. Cecilia Geijer
Chalmers tekniska hogskola AB
Department of Life Sciences, Division of Industrial Biotechnology
Kemigården 4
Gothenburg 412 96
Sweden

Re: Spectrum00245-23 (Yeasts have evolved divergent enzyme strategies to deconstruct and metabolize xylan)

Dear Dr. Cecilia Geijer:

Thank you for submitting your manuscript to Microbiology Spectrum and apologies for the late response. We have now (finally) received two reviewer opinions of your manuscript. Both reviewers find your work very nice, well written, and worthy of publication, but also make a few suggestions for improvements.

Link Not Available

Sincerely,

Florian Freimoser

Journals Department
Reviewer comments:

Reviewer #1 (Comments for the Author):

Microbiology Spectrum - Spectrum00245-23

Major Revisions

Line 110 - 125; and Figure 2G - 2K - It would strengthen the article if statistical analysis could be used to determine whether the

localizations of the different enzymes (secreted, surface-associated or intracellular) are significantly different. For example, 21 for *B. mokoensis* the error bars for cell and intra overlap. If the statistical analysis impacts the results and discussions, changes should be made throughout.

Line 137-139: The authors note that the co-culture resulted in higher OD than the mono-culture. However, the error bars at 120 h for the Bm+Wc (1:1) extends nearly to the *B. mokoensis* growth curve. Therefore, some statistical analysis here should be applied. If the statistical analysis impacts the results and discussions, changes should be made throughout.

Line 155 - 166 - It would be very helpful for reader comprehension to include this information in a table - ie) species, enzyme family, codon optimized gene, etc. - in the Results section

Figure 6 legend - Using A i), ii), iii), iv) - B i), ii), iii), iv) etc) is likely better than second from left, second from right

Line 352 - 354 - "These may be more successfully transferred to a yeast host such as *S. cerevisiae* than the enzymes from filamentous fungi and bacteria used today." - In this manuscript the authors describe that the recombinant xylanolytic enzymes were expressed in *P. pastoris*, purified using SDS-page and assayed for linearity of activity in xylan. Given the proposed industrial application of this research and the data the authors have generated, it would be helpful for them to briefly compare their data for recombinant enzyme preparations with similar reports in the literature to give this commentary more strength.

Minor Revisions

Line 85 - Specify "Isolated from the gut of wood-boring insects....."

Line 87 - Specify ".... isolated from Canadian Red pine"

Line 90 - If BmGH11 and SlGH10 are only used once, this shorthand notation is likely not necessary. Suggest writing out the genus and species name for Bm and Sl in full

Line 121-122 - "Only *B. mokoensis* showed significant" - again, would be good to specify "statistical significance"

Line 130 - ".....growth profiles and the enzyme assays" - specify Figure 2

Line 355 - Use "novel" in place of new to cut down on repetition

Line 121-122 - "Only *B. mokoensis* showed significant" - again, would be good to specify "statistical significance"

Line 130 - ".....growth profiles and the enzyme assays" - specify Figure 2

Line 355 - Use "novel" in place of new to cut down on repetition

Reviewer #2 (Comments for the Author):

The manuscript presents nice new data on three ascomycetous yeasts from diverse environments and belonging to different phylogenetic clades on growth on xylan, and their xylanolytic enzyme systems. Indeed, quite little is still known on how yeasts break down and metabolize xylan. Three selected yeasts, *Blastobotrys mokoensis* (from soil), *Scheffersomyces lignosus* (from insect guts) and *Wickerhamomyces canadensis* (from trees) all grew on xylose but differed in their ability to grow on polymeric xylan and xylan degrading enzyme system. *W. canadensis* was not growing on xylan but interestingly presence of xylooligosaccharides (XOS) or co-culturing with *B. mokoensis* activated xylan utilization machinery. Furthermore *W. canadensis* lacked obvious GH10 and GH11 family xylanase candidates but has putative xylanases from GH5. Indeed, GH5_49 from *W. canadensis* was shown to be a xylanase. This is the first report on xylanase activity in this GH5 subfamily. Manuscript also shows, that GH5_49 appears enriched in yeast compared to filamentous fungi and bacteria.

The manuscript is nicely written and easy to follow.

My critical comment is on interpretation of xylan degradation products by HPAEC-PAD. Unfortunately no chromatograms are shown. Hydrolysis of arabinoxylan (AX) produces in addition to linear XOS a series of arabinoxylooligosaccharides (AXOS), which may elute close to longer linear XOS (identified in the study as X4 - X6). This needs to be taken into account when interpreting the HPAEC-PAD results. Some AXOS are also commercially available for standards. Hydrolysis on glucuronoxyylan (GX) analogously produces MeGlcA-XOS.

The second critical comment is on Figure 7. It is overinterpreting presented data, as only endoxylanases were biochemically studied in this work, thus what other true xylan-acting enzymes these three yeast have, is still quite speculative, especially as only model substrates were used to assay the accessory activities. In addition, many of the enzymes are esterases, which were little discussed in the text, and deacetylated xylans (AX, GX) were used as substrates for xylanase characterization.

Staff Comments:

Preparing Revision Guidelines

Please return the manuscript within 60 days; if you cannot complete the modification within this time period, please contact me. If you do not wish to modify the manuscript and prefer to submit it to another journal, please notify me of your decision immediately so that the manuscript may be formally withdrawn from consideration by Microbiology Spectrum.

Microbiology Spectrum - Spectrum00245-23

Major Revisions

Line 110 - 125; and Figure 2G - 2K - It would strengthen the article if statistical analysis could be used to determine whether the localizations of the different enzymes (secreted, surface-associated or intracellular) are significantly different. For example, 2I for *B. mokoennaii* the error bars for cell and intra overlap. If the statistical analysis impacts the results and discussions, changes should be made throughout.

Line 137-139: The authors note that the co-culture resulted in higher OD than the mono-culture. However, the error bars at 120 h for the Bm+Wc (1:1) extends nearly to the *B. moekenaii* growth curve. Therefore, some statistical analysis here should be applied. If the statistical analysis impacts the results and discussions, changes should be made throughout.

Line 155 - 166 - It would be very helpful for reader comprehension to include this information in a table - ie) species, enzyme family, codon optimized gene, etc. - in the Results section

Figure 6 legend - Using A i), ii), iii), iv) - B i), ii), iii), iv) etc) is likely better than second from left, second from right

Line 352 - 354 - "*These may be more successfully transferred to a yeast host such as S. cerevisiae than the enzymes from filamentous fungi and bacteria used today.*" - In this manuscript the authors describe that the recombinant xylanolytic enzymes were expressed in *P. pastoris*, purified using SDS-page and assayed for linearity of activity in xylan. Given the proposed industrial application of this research and the data the authors have generated, it would be helpful for them to briefly compare their data for recombinant enzyme preparations with similar reports in the literature to give this commentary more strength.

Minor Revisions

Line 85 - Specify "Isolated from the gut of wood-boring insects...."

Line 87 - Specify "... isolated from Canadian Red pine"

Line 90 - If BmGH11 and SlGH10 are only used once, this shorthand notation is likely not necessary. Suggest writing out the genus and species name for Bm and Sl in full

Line 121-122 - "Only *B. mokoennaii* showed significant..." - again, would be good to specify "statistical significance"

Line 130 - ".....growth profiles and the enzyme assays" - specify Figure 2

Line 355 - Use "novel" in place of new to cut down on repetition

Dear Dr. Freimoser,

We would like to thank you for the positive feedback on our manuscript (Spectrum00245-23, Yeasts have evolved divergent enzyme strategies to deconstruct and metabolize xylan), and for forwarding the useful comments from the reviewers. Our answers to each comment can be found below, in red text where line numbers refer to the revised manuscript, and we provide both the marked-up as well as a new clean version of the manuscript in our resubmission. We have also addressed a few small typos in the text. We hope that with these edits, our manuscript will be seen as worthy for publication in Microbiology Spectrum.

On behalf of the authors,
Cecilia Geijer

Response to reviewers

Reviewer comments:

Reviewer #1 (Comments for the Author):

Microbiology Spectrum - Spectrum00245-23

Major Revisions

Line 110 - 125; and Figure 2G - 2K - It would strengthen the article if statistical analysis could be used to determine whether the localizations of the different enzymes (secreted, surface-associated or intracellular) are significantly different. For example, 2I for *B. mokoensis* the error bars for cell and intra overlap. If the statistical analysis impacts the results and discussions, changes should be made throughout.

It is a valid suggestion to include statistical analysis where appropriate for comparisons. However, the activity data presented in Figure 2 cannot easily be compared in this fashion. These experiments were, as written in the figure title and corresponding text, designed to monitor the localization of activity, and comparing cell-attached activity to secretome- or intracellular activities is not straightforward as we cannot readily determine for instance the protein amount on the cell surface or the total surface area of the cells without major guesswork. We have included a few more details in the method procedure on subcellular localization activities to clarify the amounts of cells assayed ($OD_{600}=0.25$) (line 399-405). We have also included a statistical comparison between the cells of *W. canadensis* exposed to only xylan or XOs+xylan, as these represent comparable conditions. Here we see a clear significant difference ($p<4.5E-6$) for the secreted xylanase activity and around the cell ($p<3.3E-5$). This aligns well with the observation that the species is growing on xylan only if it is exposed to XOs, which we assume trigger its xylanolytic machinery. We used one way ANOVA with Tukey test to compare groups in Origin Pro 2020. We have adjusted the text to focus more clearly on where we find activity or not, and we also state that the 'accessory' activities (xylosidase, arabinofuranosidase, glucuronidase, esterase) were comparatively weak in these conditions (lines 118-123).

Line 137-139: The authors note that the co-culture resulted in higher OD than the mono-culture. However, the error bars at 120 h for the Bm+Wc (1:1) extends nearly to the *B. mokoensis* growth curve. Therefore, some statistical analysis here should be applied. If the statistical analysis impacts the results and discussions, changes should be made throughout.

We have included a statistical analysis using one way ANOVA with Tukey test for individual timepoints of *Bm* mono- and co-culture with *Wc*. There were significant differences during the earlier growth stages (P values between 0.05 and 0.01) between the mono-culture and the co-cultures, which are now indicated in the figure and legend. The main text has been adjusted to the results of the statistical analysis on lines 140-41. At the later timepoints there were however no significant differences at 120 h between *Bm* alone and *Bm:Wc* 1:1 nor *Bm:Wc* 1:10, as expected from the reviewer.

Line 155 - 166 - It would be very helpful for reader comprehension to include this information in a table - ie) species, enzyme family, codon optimized gene, etc. - in the Results section

Thanks for the suggestion. We have now supplied a new Table 1 in line 173 with the targeted enzyme families from each yeast species. Gene sequences can be found in the supplemental list S1.

Figure 6 legend - Using A i), ii), iii), iv) - B i), ii), iii), iv) etc) is likely better than second from left, second from right

Indeed, this will help clarifying the figure. We have now included the suggested i), ii), iii), iv) for each enzyme structure and the figure legend has been adjusted accordingly as well as the result section (line 259-274).

Line 352 - 354 - "These may be more successfully transferred to a yeast host such as *S. cerevisiae* than the enzymes from filamentous fungi and bacteria used today." - In this manuscript the authors describe that the recombinant xylanolytic enzymes were expressed in *P. pastoris*, purified using SDS-page and assayed for linearity of activity in xylan. Given the proposed industrial application of this research and the data the authors have generated, it would be helpful for them to briefly compare their data for recombinant enzyme preparations with similar reports in the literature to give this commentary more strength.

The purpose of our study was not to optimize the production of the chosen enzymes but rather to study their activities and possible roles. In future studies, it would of course be interesting to study whether these enzymes compare to others in terms of yields in different species, but that also requires optimization work regarding protein production specifically. The sentence has been clarified in the Discussion to reflect our focus, and now reads:

"Understanding these strategies may be highly useful to transfer such capabilities to an industrial yeast host such as *S. cerevisiae*, which has a similar yeast morphology and might benefit from cell-attached CAZymes, compared to enzymes from filamentous fungi and bacteria that are typically secreted as soluble enzymes"

Minor Revisions

Line 85 – Specify "Isolated from the gut of wood-boring insects"
Included.

Line 87 – Specify "... isolated from Canadian Red pine"

Included.

Line 90 – If BmGH11 and SlGH10 are only used once, this shorthand notation is likely not necessary. Suggest writing out the genus and species name for Bm and Sl in full

Corrected.

Line 121-122 – “Only *B. mokoena*i showed significant.....” – again, would be good to specify “statistical significance”

See earlier response.

Line 130 – “.....growth profiles and the enzyme assays” – specify Figure 2

This has been included.

Line 355 – Use “novel” in place of new to cut down on repetition

New changed to novel.

Reviewer #2 (Comments for the Author):

The manuscript presents nice new data on three ascomycetous yeasts from diverse environments and belonging to different phylogenetic clades on growth on xylan, and their xylanolytic enzyme systems. Indeed, quite little is still known on how yeasts break down and metabolize xylan. Three selected yeasts, *Blastobotrys mokoena*i (from soil), *Scheffersomyces lignosus* (from insect guts) and *Wickerhamomyces canadensis* (from trees) all grew on xylose but differed in their ability to grow on polymeric xylan and xylan degrading enzyme system. *W. canadensis* was not growing on xylan but interestingly presence of xylooligosaccharides (XOS) or co-culturing with *B. mokoena*i activated xylan utilization machinery. Furthermore *W. canadensis* lacked obvious GH10 and GH11 family xylanase candidates but has putative xylanases from GH5. Indeed, GH5_49 from *W. canadensis* was shown to be a xylanase. This is the first report on xylanase activity in this GH5 subfamily. Manuscript also shows, that GH5_49 appears enriched in yeast compared to filamentous fungi and bacteria.

The manuscript is nicely written and easy to follow.

Thank you for the positive feedback!

My critical comment is on interpretation of xylan degradation products by HPAEC-PAD. Unfortunately no chromatograms are shown. Hydrolysis of arabinoxylan (AX) produces in addition to linear XOS a series of arabinoxylooligosaccharides (AXOS), which may elute close to longer linear XOS (identified in the study as X4 – X6). This needs to be taken into account when interpreting the HPAEC-PAD results. Some AXOS are also commercially available for standards. Hydrolysis on glucuronoxylan (GX) analogously produces MeGlcA-XOS.

We agree that this information is very useful to include, and we have now provided chromatograms for beechwood glucuronoxylan and wheat arabinoxylan hydrolysis for each enzyme in the new supplemental figure S5 (and updated the figure numbering accordingly). As can be seen, there are some overlapping peaks especially around X3, though most peaks appear at later retention time (19-25 min) and likely represent larger and branched XOs or AXOs, though standards to quantify these long XOs and AXOs were not available to us. Furthermore, Megazyme who used to sell a broader selection of branched GX- and AX-derived oligosaccharides has discontinued most of these products. For the overlapping peaks at X3, we have included a clarification in the text that these overlapping

peaks were seen, that were shown also in the previous submission as “X3-?”. We hope that the data presented using mainly quantification of the linear X1-X6 are sufficient, as in many previous xylanase studies. We have included clarifying text, lines 217-221:

“Around X3, overlapping peaks were observed especially from GX hydrolysis (Fig. S5), which likely represent branched oligosaccharides for which we did not have corresponding standards, but these were tentatively quantified using the X3 standard (Fig. 4). Larger XOs (>X6), of which many are likely branched, were also observed later in the HPAEC-PAD chromatograms (Fig. S5).”

We noticed also a smaller mistake in the reported XOs values for wheat arabinoxylan for xylopentaose (X5) and xylohexaose (X6) for WcXyn5_49A and WcXyn5_22A, where the actual values were smaller than in the previous submission, and we have adjusted the figure. This correction does not affect our overall results since these XO amounts were very low (and now are practically not visible in the chart), but we apologize for this mistake.

The second critical comment is on Figure 7. It is overinterpreting presented data, as only endoxylanases were biochemically studied in this work, thus what other true xylan-acting enzymes these three yeast have, is still quite speculative, especially as only model substrates were used to assay the accessory activities. In addition, many of the enzymes are esterases, which were little discussed in the text, and deacetylated xylans (AX, GX) were used as substrates for xylanase characterization.

We agree that the figure is speculative, and it is possible that other/unknown endo-xylanases may be present in these yeasts which were not characterized here. However, we believe that Figure 7 presents a good overview to discuss the biological role of each yeast. The biochemically studied enzymes from this study are shown clearly as 3D protein structures, while “speculative” dbCAN2 predicted CAZymes with indicative activity from the pNP substrates are marked in grey. We have added some text in the legend to further clarify that the exact enzymes involved in the process are not fully known: “Putative enzymes, as indicated from activity on chromogenic model substrates, are shown in gray together with assumed CAZy family memberships based on previous work (10)”.

In our previous study (Ravn 2021), we showed that these yeasts only possess CEs from families CE1 and CE4, which we have clarified also in the main text with “The detected esterase activity in all yeasts is attributed to CEs from families CE1 or CE4, as these are the only ones found in these species (10). More detailed studies on acetylated xylan or XOs would however be needed to conclusively demonstrate acetyl xylan esterase or potentially feruloyl esterase activity.” in line 321-325.

We have removed the word “Interestingly,” in line 284 and 316.

April 8, 2023

Dr. Cecilia Geijer
Chalmers tekniska hogskola AB
Department of Life Sciences, Division of Industrial Biotechnology
Kemigården 4
Gothenburg 412 96
Sweden

Re: Spectrum00245-23R1 (Yeasts have evolved divergent enzyme strategies to deconstruct and metabolize xylan)

Dear Dr. Cecilia Geijer:

Your manuscript has been accepted, and I am forwarding it to the ASM Journals Department for publication. You will be notified when your proofs are ready to be viewed.

Sincerely,

Florian Freimoser
Editor, Microbiology Spectrum
